# Superior normalization using total protein for western blot analysis of human adipocytes

**Leo J.S. Westerberg**[1]ⓔ, **Benjamin Dedic**[1]ⓔ, **Erik Näslund**[2], **Anders Thorell**[3],
**Kirsty L. Spalding**[1]*

**1** Department of Cell and Molecular Biology, Karolinska Institutet, Stockholm, Sweden, **2** Department of Clinical Sciences, Danderyd Hospital, Karolinska Institutet, Stockholm, Sweden, **3** Department of Clinical Sciences, Danderyd Hospital, Karolinska Institutet and Department of Surgery, Ersta Hospital, Karolinska Institutet, Stockholm, Sweden

ⓔ These authors contributed equally to this work.
* kirsty.spalding@ki.se

## Abstract

Western blotting has been a pivotal method for analyzing protein expression since the late 1970s, yet there is no established consensus on an optimal normalization strategy. In this study, primary mature human adipocytes were used to investigate the robustness of housekeeping proteins and total protein (TP) as normalization references for western blotting. TP exhibited the lowest variance among technical replicates compared to all investigated housekeeping proteins and was a superior normalization reference for the chosen protein-of-interest. TP also demonstrated the closest alignment with expected values when loaded as a protein gradient, highlighting the dynamic strength of TP as a normalization standard. Additionally, TP consistently demonstrated lower intra- and inter-individual variability in comparison to housekeeping proteins investigated across three metabolically similar individuals. In conclusion, TP normalization is the preferred method for reliable protein expression analysis in primary mature human adipocytes.

## Introduction

Western blotting is a widely used technique for the semi-quantitative assessment of protein levels. The method was first developed in the late 1970s [1,2] and along with improvements remains relevant today. While many variations of the method have been established, the fundamentals persist: Proteins are separated by size using electrophoresis and subsequently quantified using antigen-specific reagents. As antigen detection correlates with the amount of protein loaded, normalization is implemented to correct for variations between loaded samples, thereby enhancing the accuracy of comparisons across multiple samples.

Historically, housekeeping proteins such as glyceraldehyde 3-phosphate dehydrogenase (GAPDH), actin, and tubulin have been used to control for loading and

**Data availability statement:** All relevant data are within the paper and its Supporting Information files.

**Funding:** This work was supported by grants to K.L.S. from the Swedish Research Council (2022-01236, https://www.vr.se), the Strategic Research Program for Diabetes at Karolinska Institutet (C5471162, https://ki.se/en/srp-diabetes), the Novo Nordisk Foundation (C5475033, https://novonordiskfonden.dk/en), Vallee Foundation Scholar Award (C5471234, https://www.thevalleefoundation.org), Mark Foundation Aspire Grant (C5477023, https://themarkfoundation.org), The Swedish Cancer Society (22 2420, https://www.cancerfonden.se), the Strategic Research Program for Stem cells and Regeneration at Karolinska Institutet (C5472022, https://ki.se/en/research/research-areas-centres-and-networks/strategic-research-areas/stratregen-strategic-research-area-in-stem-cells-and-regenerative-medicine), and Knut and Alice Wallenberg Foundation (2020.0118, https://kaw.wallenberg.org/en). AT was supported by the Erling-Persson Foundation (2022 0163, https://erlingperssons-stiftelse.se). The funders had no role in study design, data collection and analysis, decision to publish, or preparation of the manuscript.

**Competing interests:** The authors have declared that no competing interests exist.

transfer inconsistencies. These proteins are considered housekeeping proteins because they are fundamental to normal cell function and are thus constitutively expressed across all cells and tissues. Consequently, these housekeeping proteins can be quantified and utilized as normalization references to accurately compare proteins of interest across multiple samples. Numerous studies, however, have demonstrated that housekeeping proteins are variable, both across cell lines [3]–[5] and various human tissues, such as the human cortex [6], cerebrospinal fluid [7], skeletal muscle [8] and human placenta [9]. Whilst housekeeping proteins have been studied for their reliability in adipose tissue [10], we investigate their applicability for studies in isolated adipocytes and compare normalization using housekeeping proteins to an alternative method, namely total protein (TP) normalization.

In contrast to the use of specific housekeeping proteins, TP utilizes dyes, or other chemistry, to produce a signal that represents the entire protein content of a sample. Building on this approach, TP normalization effectively eliminates the drawbacks of individual protein expression variability and more accurately reflects the true sample load [11]–[13]. Common staining methods for TP include Coomassie Brilliant Blue, Ponceau Red, Amido Black and SYPRO Ruby staining. Despite their effectiveness in reflecting TP content, the above-mentioned TP methods introduce additional wash and de-staining steps, extending the already lengthy process of western blotting. To maintain the advantages of TP normalization while simplifying the process, stain-free technology was developed. Stain-free technology utilizes a trihalo compound that can be incorporated into electrophoresis gels where they covalently bind to tryptophan residues upon UV activation. This creates a complex that emits fluorescent light directly proportional to the number of complexes [14], enabling quick TP detection without washes or de-staining.

Despite these established advantages, TP normalization has not yet achieved universal adoption in western blot analysis. The persistence of traditional housekeeping protein methods can be attributed to several factors, i) historical precedent has maintained housekeeping proteins as the default choice in many laboratories, particularly given their extensive representation in published literature, ii) early iterations of TP detection methods introduced technical complexities and potential interference with subsequent immunodetection steps and iii) the requirement for specialized imaging equipment and standardized detection protocols has posed implementation barriers. Moreover, the diverse array of TP detection methods, from classical chromogenic stains to stain-free technologies, has limited the establishment of a unified normalization protocol.

Considering the advantages of TP normalization, this study investigates if stain-free based TP normalization is a more suitable alternative to traditional housekeeping protein normalization in western blotting of primary mature human adipocytes.

## Materials and methods

### Ethics

All experiments were performed in accordance with the statutes of the Declaration of Helsinki. The study was approved by the regional ethics committee in Stockholm

(Regionala etikprövningsnämnden i Stockholm, 2014/1115–31 and 2017/1156-32). Sample collection for this study began on 9th of September 2021 and continued through 13th of March 2023 at both Ersta Hospital and Danderyd Hospital, Stockholm, Sweden. Informed written consent was obtained from all participants prior to surgery.

### Human samples

Human surgical biopsies from omental (OM) and abdominal subcutaneous (SC) white adipose tissue were obtained from 9 individuals with obesity undergoing bariatric surgery at Ersta Hospital (n = 6) or Danderyd Hospital (n = 3) in Stockholm, Sweden. Individuals were fasted overnight and blood glucose, plasma insulin and c-peptide were measured either the morning of surgery (Ersta Hospital) or 2 weeks prior to surgery (Danderyd Hospital). Bloodwork was analyzed at the Karolinska Universitetslaboratoriet. A summary of clinical and metabolic parameters of the full cohort can be found in S3 and S4 Tables.

### Adipocyte isolation

Adipocytes were extracted from the tissue sample using the method outlined in [15], with the only difference being the exclusion of Bovine Serum Albumin (BSA). In brief, the tissue sample was finely minced and subjected to digestion in Krebs-Ringer-phosphate buffer (containing 127 mM NaCl, 12.3 mM NaPO4, 1.36 mM CaCl2, 5.07 mM KCl, 1.27 MgSO4, pH 7.4) that had 0.05% collagenase I from Clostridium histolyticum (Sigma, #C0130), 5 mM D-glucose and 50 µg/ml gentamicin. Digestion took place in a 37°C oscillating water bath for 45−60 minutes. Residual undigested tissue was filtered out using a 250 µm nylon filter. Adipocytes were washed three times with a wash buffer comprising of 5 mM D-glucose and 50 µg/ml gentamicin in DPBS (Gibco, #14190−136) at pH 7.4. After the final wash, the cells were transferred to 2 mL tubes, rapidly frozen in liquid nitrogen and stored at −80°C for subsequent protein extraction.

### Protein extraction and quantification

Frozen adipocytes were mixed at 2:1 with 1% SDS buffer (Sigma, #05030–500ML-F) supplemented with Halt™ Protease & Phosphatase Single-Use Inhibitor Cocktail (100X) (Thermo Scientific, 1:100, #78442). The mixture was heated at 95°C for 5 minutes using a heating block, followed by incubation on a rocker at room temperature for 15 minutes. Samples were centrifuged at 14.000 rcf for 10 minutes and the aqueous phase transferred to a new tube. Protein concentrations were determined using a Pierce™ BCA Protein Assay (Thermo Scientific, #23227). Absorption was measured using FLUOstar OPTIMA spectrophotometer and protein concentrations calculated using a linear regression model fitted to the BSA standard (125–2000 ng/µL), accounting for blanks.

### Western blot

For each sample, 10 µg of protein was mixed with 4x Laemmli buffer (BioRad, #1610747) supplemented with 2-Mercaptoethanol (BioRad, #1610710) at 1:3 in 0.5 mL tubes. Samples were heated in a heating block at 95 °C for 5 min and subsequently centrifuged at 20817 rcf for 1 min. Running buffer was prepared by mixing 100mL of 10x TGS stock solution (BioRad, #1610772) with 900 mL ddH₂O. Precast Mini-PROTEAN TGX Stain-Free Gels (BioRad, #4568094) were used for gel electrophoresis. 3 µL of Precision Plus Protein Unstained Standards (BioRad, #1610363) was loaded in the far left well and protein samples in the remaining wells. Gel electrophoresis was run at 200V for approximately 40 min with a PowerPac™ Basic (BioRad, #1645050).

After electrophoresis, the gel was activated in the ChemiDoc™ MP Imaging System (BioRad, #12003154) using stain-free technology. Prior to assembling the transfer stack, a low fluorescence (LF) PVDF membrane was activated in pure methanol (Sigma-aldrich, #32213-1L-M) and filter papers soaked in 4°C 1x transfer buffer (Trans-Blot Turbo RTA Transfer Kit LF PVDF, BioRad, #1704274). A transfer stack was constructed in the Trans Blot® Turbo™ Transfer system (BioRad) and a mixed molecular weight protein transfer program was run for 7 min. The membrane was imaged in the BioRad

ChemiDoc MP imaging system using stain-free UV activation for TP quantification as well as to ensure successful protein transfer had occurred.

## Immunoblotting

The membrane was blocked in EveryBlot Blocking Buffer (BioRad, #12010020) for 10 min on a shaker at RT. After blocking, the membrane was incubated with primary antibodies O/N on a shaker at 4°C. Primary antibodies, all commercially validated, were prepared in 10 mL 5% milk (Millipore, #70166-500G) at the following concentrations; BSA (Life, 1:1000, #A11133), GLUT4 (ProteinTech, 1:500, #66846–1-Ig), PARK7 (Abcam, 1:500, #ab18257), ENOA (MyBioSource, 1:250, #MBS9231722) and FAH (Invitrogen, 1:500, #PA5−42049). After primary antibody incubation, the membrane was washed 3 times in TBS supplemented with 0.05% tween-20 (TBST) (ChemCruz, #SC-362311) and incubated with secondary antibody for 1 hour at RT. Secondary antibodies and GAPDH, actin or tubulin were prepared in 10 mL 5% milk (in PBS) at the following concentrations; Anti-Mouse IgG H&L HRP conjugated (Abcam, 1:5000, ab205719), hFAB Rhodamine Anti-GAPDH (BioRad, 1:1250, #12004167), hFAB Rhodamine Anti-Actin (BioRad, 1:1250, #12004163), hFAB Rhodamine Anti-Tubulin (BioRad, 1:2500, #12004165) and depending on the housekeeping used, StarBright Blue 700 Goat Anti-Rabbit IgG (BioRad, #12004161) diluted either at 1:2500 (PARK7 and FAH) or 1:5000 (ENOA). The membrane was washed twice in TBST and once in DPBS (Gibco, #14190−136) and imaged in the ChemiDoc MP imaging system. Housekeeping proteins were imaged at 700nm (PARK7, ENOA or FAH), followed by rhodamine conjugated housekeeping proteins, GAPDH, actin or tubulin. The membrane was incubated in 5 mL SuperSignalTM West Femto Maximum Sensitivity Substrate (Thermo Scientific, #34096) ECL mix for 5 min on a shaker at RT. Lastly, GLUT4 was imaged using chemiluminescence.

## Western blot analysis

The volume tool in ImageLab software (BioRad) was used to analyze stain-free images of the membranes, taken immediately post transfer. Housekeeping proteins and GLUT4 band intensities were determined utilizing the lane and band tool in ImageLab. Visualization of data and statistics were performed using Prism 9 software (GraphPad).

## Western blot normalization

In fixed point normalization, the choice of reference point can introduce a bias [16]. If the selected reference point does not accurately represent the tendency or the range of the data, it can skew the normalized values. This bias can lead to misinterpretation of results, as the normalized data might disproportionately emphasize certain values while under representing others.

To avoid fixed point normalization bias, data was normalized by first dividing the intensity of each band/lane by the total intensity of all bands/lanes within each blot (in Equation 1 shown below, denoted as the 'observed fraction of total intensity'). This fraction was then divided by the expected ratio (denoted in Equation 2 as the 'expected fraction of total intensity'), i.e., the theoretical ratio obtained by dividing the intensity of a given band by the total sum of intensities for all bands/lanes (S1 Fig and S1 Table).

## Normalization calculations

1. Calculate the Fraction of Total Intensity:

$$Observed\ Fraction\ of\ Total\ Intensity = \frac{Band\ Intensity}{\Sigma\ Band\ Intensities}$$

This step determines the proportion of the total intensity contributed by each band.

2. Calculate the Deviation from Expected Intensity:

$$\text{Deviation from Expected Intensity} = \frac{\text{Observed Fraction of Total Intensity}}{\text{Expected Fraction of Total Intensity}}$$

This step compares the observed fraction of total intensity to the expected fraction, resulting in:

> 1 if the observed intensity is higher than expected
< 1 if the observed intensity is lower than expected
= 1 if the observed intensity matches the expected intensity

Following established statistical protocols, the coefficient of variation (CV) was selected to assess the measured variability (precision) of normalization references both within technical replicates [5,7,9] and across a dynamic range of protein loading [17,18] CV was calculated using the deviation values obtained from the equations above. The standard deviation was divided by the mean of the deviation values and multiplied by 100 to generate a percentage using Prism 9 software (GraphPad), quantifying the relative variability in protein expression across experimental conditions [6–10].

### Simulation of normalization reference variability

To demonstrate the universal impact of normalization reference variability on protein quantification precision, simulated western blot data was generated using seeded random number generation. A protein of interest data set with realistic technical variability (CV = 19.9%) was created using the Box-Muller transformation to generate normally distributed values around a mean of 1. Normalization reference datasets with systematically increasing coefficients of variation (5%, 15% and 30%) were similarly generated using different random seeds to ensure independence. Optimal seeds were identified through iterative searching to achieve target CVs within ±1% tolerance. Normalized protein values were calculated as the ratio of protein of interest to reference values, and CVs were determined as described above.

## Results

### Theoretical impact of normalization reference variability on protein quantification

The precision of Western blot protein quantification is fundamentally determined by the variability of the normalization reference, independent of the specific protein being analyzed. To demonstrate this principle, simulated identical protein of interest (POI) data representing realistic technical variability (CV = 19.9%) was normalized using references with systematically increasing variability ranging from 5% to 30% CV (S2 Fig).

Normalization of the POI using a stable reference (5% CV) resulted in a final CV of 16.6%, closely maintaining the precision of the original protein signal (S2 Fig). However, as reference variability increased, the precision of the normalized POI deteriorated progressively. Normalization using moderately variable references (15% CV) inflated the normalized protein variability to 30.6% CV (S2 Fig). Notably, normalization using highly variable references (30% CV) resulted in final CVs of 47.0%, representing a 2.8-fold increase in variability compared to stable reference normalization (S2 Fig, 16.6% vs 47.0%). This analysis demonstrates that normalization reference selection determines normalized protein precision regardless of the protein target (S2 Fig). The 2.8-fold difference in variability occurred with no change in the underlying protein signal, confirming that these effects represent universal analytical principles rather than protein-specific phenomena.

These simulations assume identical loading conditions with CVs reflecting only imprecision in signal readout and/or reference stability. It should be noted that simulated datasets do not incorporate biology such as potential correlations between normalization references and target proteins, which falls outside the scope of this study.

To validate this principle experimentally in primary human adipocytes, GLUT4 was selected as a representative POI to compare TP and housekeeping protein normalization performance.

## BSA contamination in adipocyte isolation and its impact on western blotting

Prior to investigating the suitability of TP normalization in primary mature human adipocytes, the presence of potential protein contaminants from the adipose tissue isolation protocol was assessed. In previous western blot analyses using mature human adipocytes, strong TP bands around 60 kDa were consistently observed, suggesting potential protein contamination (Fig 1A). Bovine serum albumin (BSA) (kDa 65), which is commonly used in adipocyte isolation protocols to enhance cell survival and overall viability, was suspected to be a contributing factor to this contamination. To investigate this, an adipose biopsy was divided in half and adipocytes were isolated either with BSA (standard protocol), or without BSA (adjusted protocol). No discernible difference in adipocyte yield was seen between standard and adjusted protocols. Following adipocyte isolation, half of the cells from each condition were immediately frozen for protein isolation, while the remaining half were subjected to three washes in PBS prior to freezing. The protein concentration for each lysate was determined using a BCA kit. Despite all samples originating from the same individual biopsy, marked variations in protein concentrations were observed across the different conditions (S2 Table). Notably, the protein concentration of the

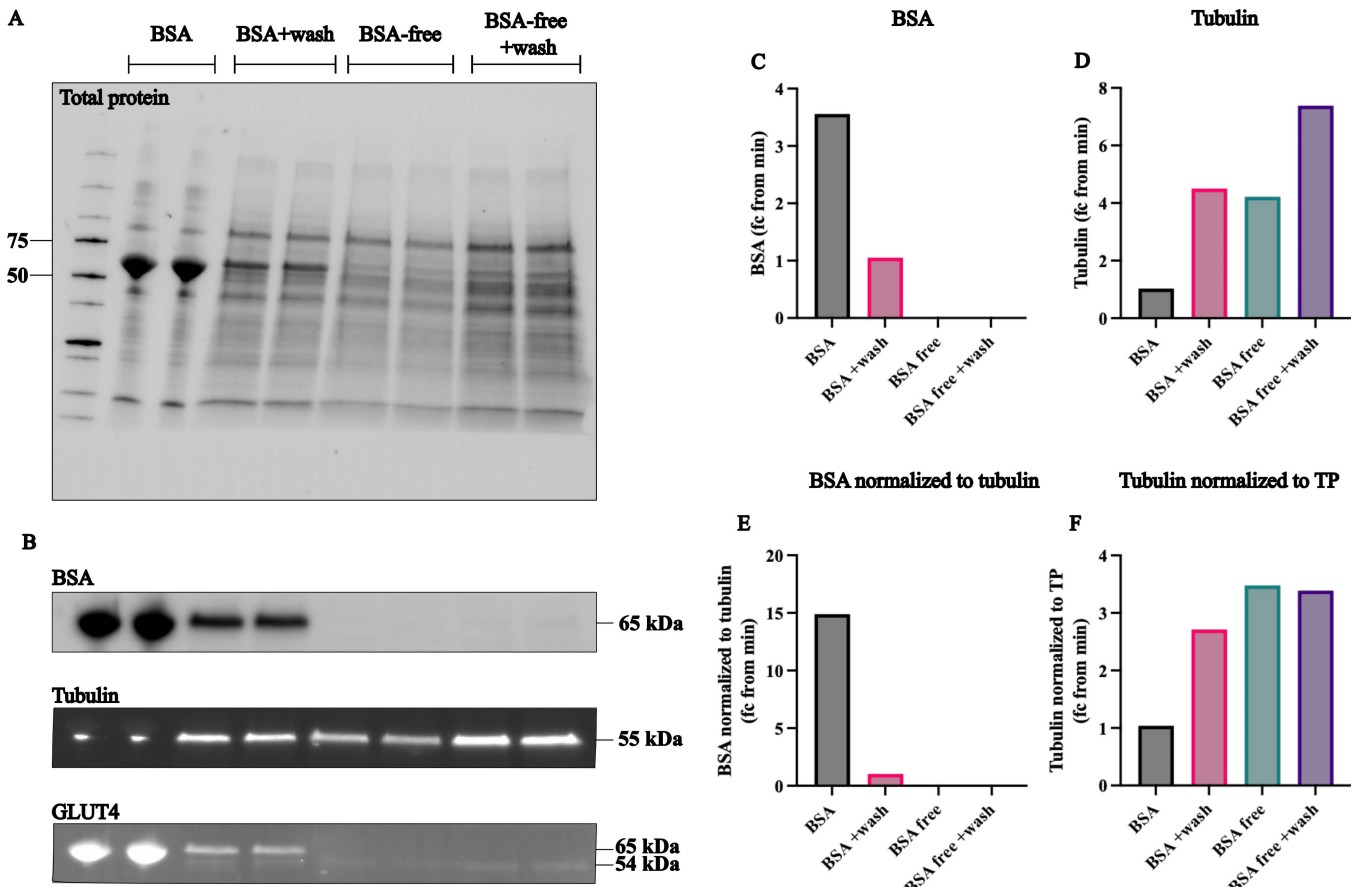

**Fig 1. Impact of BSA contamination on protein quantification methods.** (A) Stain-free image of total protein (TP) with bands corresponding to 50 and 75 kDa indicated on ladder. Duplicates shown for each condition. (B) BSA (65 kDa), tubulin (55 kDa) and GLUT4 (54 kDa) staining. (C) Average BSA signal from all duplicates, normalized to the smallest value above 0. (D) Average tubulin signal from all duplicates, normalized to the smallest value. (E) Average BSA signal from all duplicates, normalized to tubulin, normalized to smallest value above 0. (F) Tubulin normalized to TP, normalized to the smallest value.

unwashed BSA-isolated sample surpassed the upper BCA standard, leading to an extrapolated estimate of the exact concentration under linear assumptions (S3A Fig).

To visualize TP, samples were separated by electrophoresis using stain-free gels, followed by a quick UV activation step. In the unwashed BSA-isolated sample, TP was dominated by a single band in the ~60 kDa range, in contrast to a more spread-out profile in BSA-free and washed BSA-isolated samples (Fig 1A). Strong BSA labeling was seen in the ~60 kDa range in both unwashed and washed BSA-isolated adipocytes but was not detected in BSA-free samples (Fig 1B, C). The combination of reduced TP at ~60 kDa (Fig 1A) and no BSA detection through antibody labeling in the BSA-free samples (Fig 1B) indicated that BSA introduced significant protein contamination and should therefore be avoided during tissue isolation.

Additionally, BSA contamination increased TP content in the lysate, as shown by BCA readouts (S2 Table). This can lead to underloading of endogenous proteins, highlighted by tubulin measurements (Fig 1B, 1D), or in the ratio of BSA to tubulin (Fig 1E). This issue stems from the limitations of BCA to distinguish between endogenous and exogenous sources of protein. Washing BSA-isolated samples attenuates the contamination effects but does not eliminate TP normalization issues, as tubulin normalized to TP is lower in both washed and unwashed BSA-isolated samples, compared to BSA-free samples (Fig 1F).

To investigate the suitability of TP normalization compared to housekeeping proteins in isolated human adipocyte samples, glucose transporter 4 (GLUT4) was chosen as the protein of interest. GLUT4 antibody staining showed strong bands in the unwashed BSA-isolated sample, with weaker bands in the washed BSA-isolated sample (Fig 1B). Overexposure of the blot showed that the expected GLUT4 signal (~54 kDa) was clearly present in the BSA-free samples while the BSA-isolated samples showed non-specific bands at ~60 kDa (S3B Fig). These results can be explained by non-specific antibody binding to BSA (65 kDa) leading to false positives in BSA-isolated samples but not in BSA-free samples. Thus, subsequent experiments within this study used BSA-free samples, accentuating the importance of limiting BSA, or other protein contaminants, in western blotting.

## Normalization reference variability in technical replicates

To determine the optimal normalization strategy for western blot analysis in mature human adipocytes, housekeeping proteins previously validated for adipocyte and obesity research, such as parkinsonism associated deglycase (PARK7), enolase 1 (ENOA), and fumarylacetoacetase (FAH) were investigated [10]. In addition, GAPDH, actin and tubulin were included as global housekeeping markers for a broader comparison. Evaluation of all normalization references was performed using primary mature human white adipocytes. First, the assessment involved technical replicates, where each well in a gel was loaded with 10 µg of protein from the same sample. Samples were obtained from four different individuals: two obese hyperinsulinemic (OBHI) patients (one subcutaneous (SC) and one omental (OM)) and two obese normoinsulinemic (OBNI) patients (one SC and one OM). Each sample was analyzed across three separate blots to assay all housekeeping proteins, allowing for the normalization of GLUT4 levels to two housekeeping proteins and TP within each blot. Data was normalized and visualized such that an increasing deviation from 1 indicated a higher protein variation (S1 Fig, S1 Table, **see methods for more details**).

Housekeeping proteins, TP and GLUT4 were analyzed independently to investigate potential variability in loading, antibody stain or transfer efficiency. The coefficient of variation (CV) for housekeeping proteins was higher in comparison to TP (Fig 2A–F, S4A–F Fig, left-side panels, S4 Fig), except in two instances: FAH exhibited the lowest CV for the OBHI OM sample, and actin had the lowest CV for the OBNI OM sample (Fig 2D, S4F Fig). Notably, GLUT4 demonstrated high CVs prior to normalization in all samples and across all blots (Fig 2A–F, S4A–F Fig). GLUT4 was subsequently normalized to both housekeeping proteins and TP to compare the CV between each normalization reference. Unexpectedly, normalization predominantly led to an increased CV for GLUT4 (Fig 2A–F, S4A–F Fig, right-side panels). Using TP as the normalization reference resulted in the lowest CV for GLUT4 in most blots (Fig 2A, B, E, F, S4A, B, E Fig). When normalized to actin and ENOA, GLUT4 achieved the lowest CV in two blots each (Fig 2C, D, S4C, D Fig).

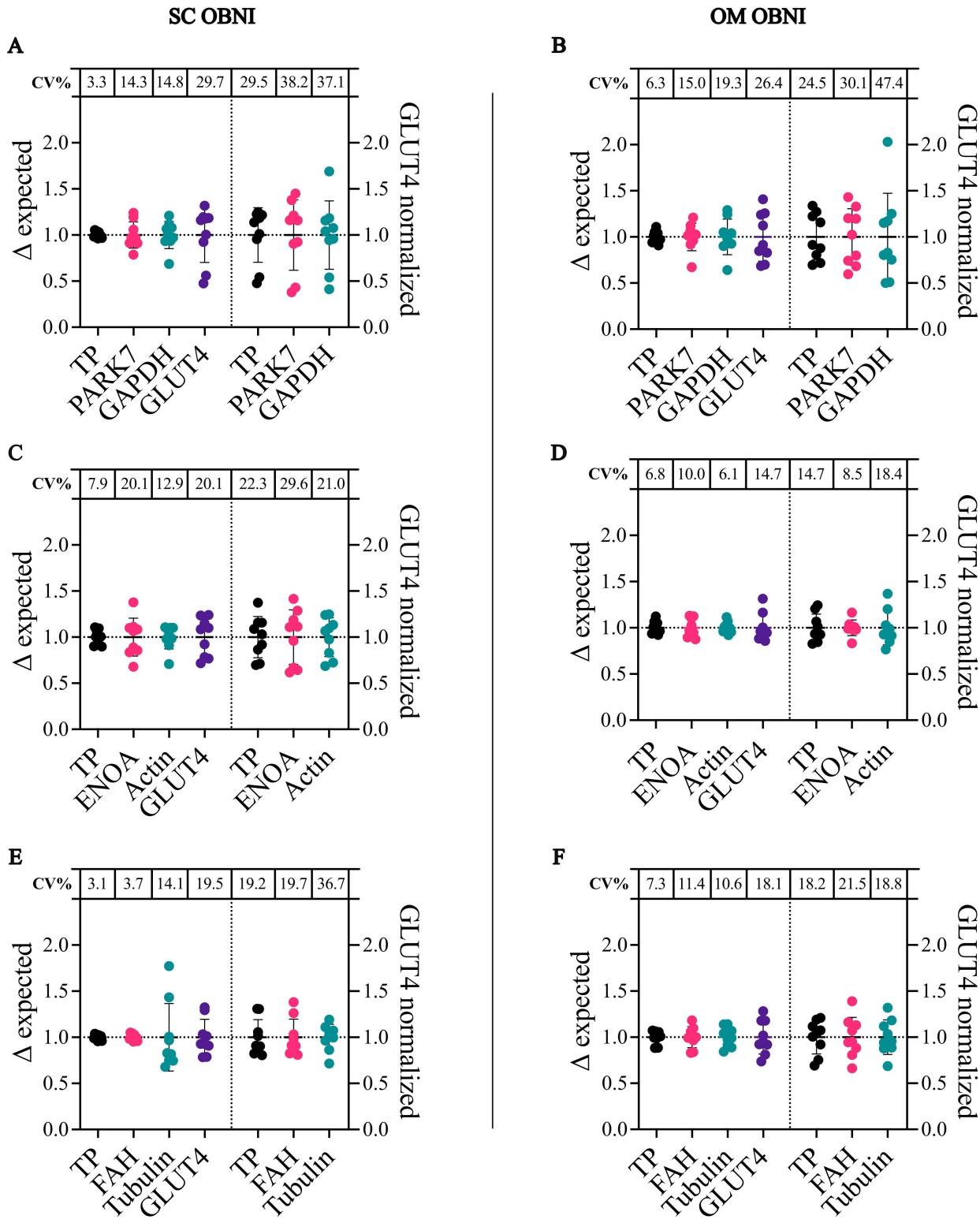

**Fig 2. Technical replicates from OBNI individuals.** Coefficient of variation (CV) for total protein (TP) and respective housekeeping proteins, normalized to the sum (left panels), and for GLUT4 normalized to TP or housekeeping proteins (right panels). Each dot represents one well – a technical replicate of the same lysate sample run in separate lanes. TP demonstrated the lowest CV across all technical replicate blots, except when compared to actin in the OM sample (A–F). When normalizing GLUT4, TP showed the lowest CV in four blots (A, B, E, and F), while actin (C) and ENOA (D) exhibited lower CVs in two blots.

The CVs resulting from GLUT4 normalization to actin and ENOA showed depot-specific trends: Actin yielded the lowest CVs in SC adipocytes (OBNI and OBHI), while ENOA had the lowest CVs in OM adipocytes (OBNI and OBHI) (Fig 2C, D, S4C, D Fig). This data suggested that actin and ENOA exhibited differing variability within adipocytes from specific adipose depots. Taken together, TP emerged as the most reliable normalization reference, offering lower variability and more consistent CVs for GLUT4 normalization across technical replicates.

## Dynamic range of normalization references in single sample loading gradient

To explore dynamic range differences between housekeeping proteins and TP, an OM adipocyte protein lysate from a individual with obesity and hyperinsulinemia was loaded with a protein gradient of 2.5–20 µg, in 2.5 µg increments (n = 8) (S6 Fig). The same normalization approach used for the technical replicates was used to visualize the data (S1 Fig and S1 Table, **see methods for more details**).

Housekeeping proteins and TP were assessed independently: In all three blots, TP exhibited the lowest CV and was the only normalization reference with a CV lower than 10% (Fig 3A–C). Lower protein loads typically resulted in larger deviations from their expected value, which in turn contributed to the overall higher CVs observed in housekeeping proteins (Fig 3A–C). Large amounts of lysate can cause signal saturation for highly expressed proteins, such as housekeeping proteins [19], as observed in our study: Signal saturation was observed in FAH where a load of a 7.5 µg and above showed a negligible increase in signal, likely contributing to the high CV (Fig 3C and S6C Fig). No signal saturation was observed in other housekeeping proteins or TP within the tested range.

GLUT4 was normalized to TP and housekeeping proteins. As observed with technical replicates, the CVs from GLUT4 normalization generally increased after normalization, regardless of the chosen reference. While GLUT4 normalization to TP yielded the lowest CV in two of three blots, there was considerable variation across all three blots (27.3 vs 15.9 vs 9.6, Fig 3D–F). This variability is primarily attributed to the CVs from GLUT4, rather than being substantially influenced by TP variability (Fig 3A–C). The signal saturation of FAH within the tested dynamic range highlighted its unsuitability as a reference protein due to the lack of linearity (Fig 3C, F, S6C Fig). A consequence of its insensitivity to protein loads from 7.5–20 µg also likely contributed to an artificially low CV in technical replicates (Fig 2E, F, S4E, F Fig).

Taken together, TP offers a more stable and reliable normalization reference compared to housekeeping proteins across a dynamic range and technical replicates. Variability in normalization with housekeeping proteins, particularly highlighted by FAH, underscores the importance of choosing a dynamic and low-variance reference protein. The relatively lower CV resulting from TP normalization, combined with no signal saturation within the tested dynamic range, suggests that TP as a more fitting normalization reference compared to housekeeping proteins.

## Comparing normalization references across samples from multiple individuals

To assess the impact of using different normalization references on GLUT4 levels across individuals, a cohort of three clinically and metabolically similar individuals was studied (S3 Table). Given the varying sensitivities observed at different protein loads during the dynamic range comparisons, samples from each individual were loaded at 5, 10 and 15 µg to minimize biasing based on dynamic range performance. Data were initially normalized to the respective protein loads (i.e., normalized such that 5 µg corresponds to a factor of 1, 10 µg to 2, and 15 µg to 3), and visualized as fold change from the minimum value. This approach was adopted since there is no predefined expected value for each sample when comparing between unique individuals.

Quantification of TP, housekeeping proteins and GLUT4 for each individual demonstrated consistent trends across the three individuals: Individual 1 showed the highest levels of protein expression for all proteins, followed by individual 3, then individual 2 (Fig 4A–C, S7 Fig). TP, ENOA, actin, FAH and GLUT4 all displayed minimal variance within individuals when normalizing to protein load (Fig 4A–C). This was similar for PARK7, GAPDH and tubulin for individuals 2 and 3, but not for individual 1 (Fig 4A–C). When comparing normalization references, TP and actin exhibited a relatively small fold-change

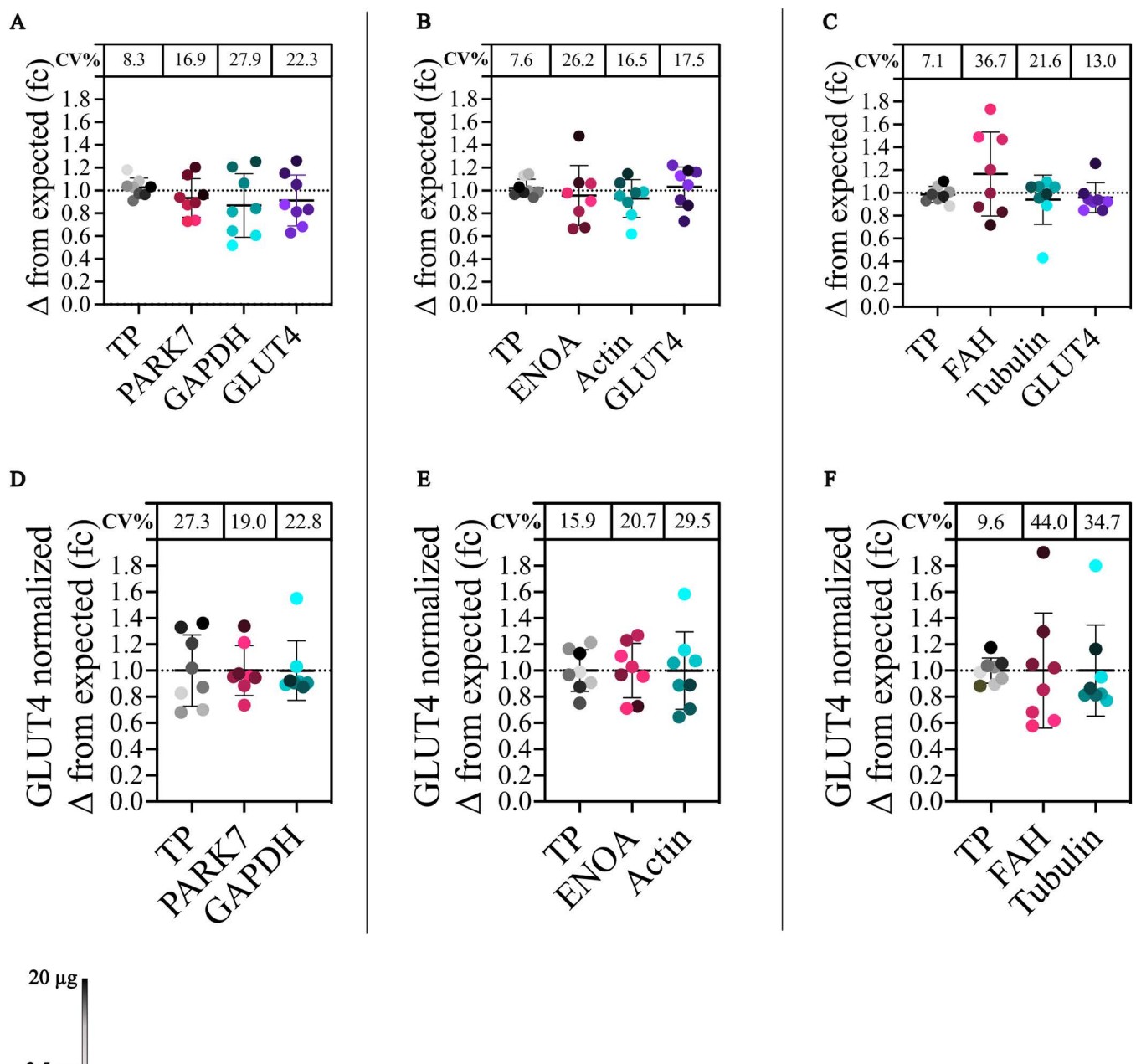

**Fig 3. OBHI OM lysate loaded in 2.5 µg gradient increments (2.5–20 µg).** Lysates were visualized with a color gradient, and the coefficient of variation (CV) was calculated. Total protein (TP) showed the lowest CV across all blots when normalized to the sum (A–C). For GLUT4 normalization, PARK7 demonstrated the lowest CV in one blot (D), while TP achieved the lowest CV in two blots (E and F).

**Mature human omental adipocyte lysates**

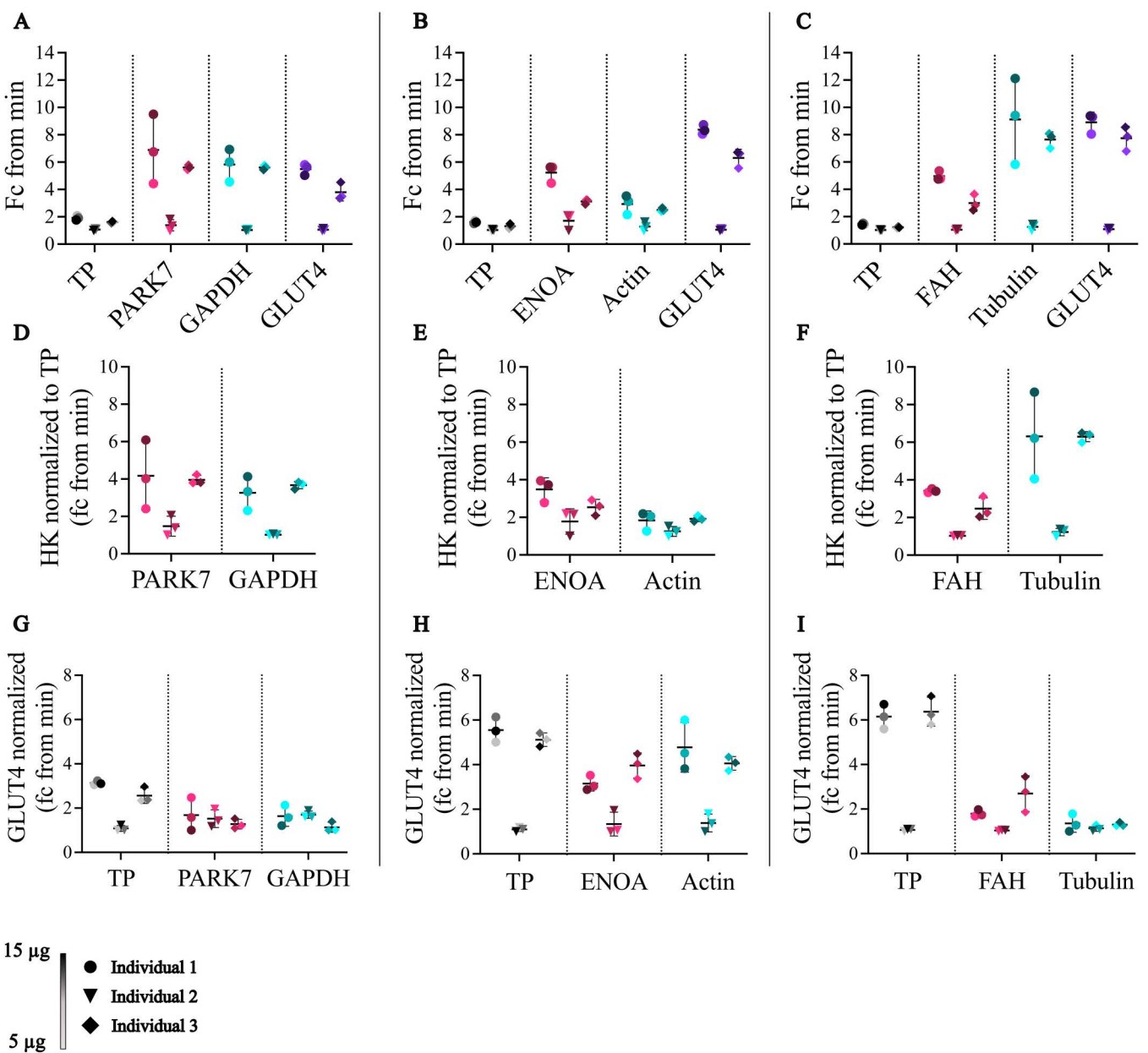

**Fig 4. Protein gradient loading across a metabolically and clinically similar cohort (n = 3).** Lysates were loaded at 5, 10, and 15 µg, visualized with a color gradient, and normalized to the minimum value. Inter- and intra-individual comparisons of normalization references and GLUT4 demonstrated that total protein (TP) exhibited the lowest variability both within and between individuals (A–C). In contrast, housekeeping proteins normalized to TP reveal substantial inter-individual variability (D–F), highlighting that their expression levels differ between patients even when loading is accounted for. GLUT4 normalized to different references (TP or housekeeping proteins) further illustrate how the choice of normalization strategy can markedly influence the apparent between-patient differences (G–I).

(fc) difference between individuals (fc~2), versus PARK7 (fc~6), GAPDH (fc~6), ENOA (fc~5), FAH (fc~4) and tubulin (fc~10) (Fig 4A–C).

To determine if variations in loading amounts were responsible for the substantial differences observed in housekeeping protein expression among individuals, the housekeeping proteins were normalized against TP. While this normalization mitigated the differences in housekeeping expression between individuals, loading amount alone could not account for these variations (Fig 4D–F), suggesting that inherent differences in housekeeping protein expression exist even amongst individuals with similar clinical and metabolic profiles.

TP normalization led to comparable expression levels of GLUT4 between individuals 1 and 3, with a notable decrease in individual 2 across all blots (Fig 4G–I). When normalized to TP, the fold change difference was approximately 6 in two blots, in contrast to a fold change difference of 3 in one blot. This variability likely stems from the major variations in GLUT4 signal, rather than from the stable TP signal observed in these individuals, with GLUT4 varying approximately 1–10-fold, and TP showing a narrower range of about 1–2-fold (Fig 4A–C). When GLUT4 is normalized against PARK7, GAPDH or tubulin, the expression level among the three individuals is consistent (Fig 4G, I). However, normalization using ENOA, actin or FAH reveals distinct difference in expression: Individual 2 consistently showed the lowest levels of GLUT4, while the highest GLUT4 expression shifts between individual 1 when normalized to actin, and individual 3 when normalized to ENOA or FAH (Fig 4H, I).

Collectively, these findings indicated that the choice of normalization reference dramatically affects the determination of GLUT4 expression levels when comparing multiple individuals. Notably, the expression of housekeeping proteins varies widely among individuals with similar clinical and metabolic profiles (Fig 4D–F), influencing the normalized values of proteins of interest (Fig 4G–I). Among the housekeeping proteins, actin demonstrated smaller inter-individual variability (Fig 4E) and yields normalized GLUT4 expression levels comparable to those obtained when normalizing to TP. Thus, normalization references with minimal intra- and inter-individual variability, such as TP and actin, are necessary for precise protein expression comparison across multiple individuals.

In summary, these findings highlight the importance of using normalization references with low variance both within and among individuals. While actin demonstrated low inter-individual variability, it under performs in both dynamic range and technical replicate analysis. Ultimately, the robust variance of TP in technical replicates, dynamic range and across individuals makes it a superior normalization reference for mature human adipocyte western blot analysis. Table 1 summarizes the performance of TP and housekeeping proteins across all experimental conditions tested, highlighting the overall best normalization reference alongside the most robust housekeeping protein alternative when experimental conditions necessitate housekeeping protein use (e.g., requirement for BSA in the isolation or culture environment).

## Discussion

Western blotting is one of the most widely utilized techniques for protein expression analysis. Despite its popularity, a consensus on the most effective normalization strategy has not been reached. Housekeeping protein normalization was adopted to standardize western blot results. As increasingly sophisticated tools and software for precise protein quantification have become available, TP normalization has emerged as an alternative approach to housekeeping protein

**Table 1. Optimal normalization references for different experimental designs.**

| Experimental setup | Best overall | Best housekeeping protein |
|---|---|---|
| Technical replicates SC | TP | FAH |
| Technical replicates OM | TP | Actin |
| Dynamic range | TP | Actin |
| Inter-individual comparison | TP | Actin |

normalization. In this study, we compare these two normalization methods in primary mature human adipocyte western blotting, to determine which normalization strategy is superior.

In our study, TP exhibited lower variance compared to both GLUT4 and housekeeping proteins across technical replicate- and dynamic range analyses. The high variance observed in GLUT4 and housekeeping proteins occasionally made normalizing GLUT4 to housekeeping proteins, rather than TP, appear advantageous. However, this apparent co-variance seemed to occur by chance, as GLUT4 and housekeeping proteins sometimes varied concurrently on one blot but independently or in opposing directions on another. Moreover, even minor fluctuations in a housekeeping protein's expression could significantly skew the normalization process, whereas similar variations in a single protein would have a negligible impact on TP normalization.

Previous research identified PARK7, ENOA, and FAH as suitable housekeeping proteins for western blotting in adipose tissue [10]. That study used two-dimensional difference gel electrophoresis to assess protein stability between SC and OM adipose tissue across individuals. In contrast, our study focused on the stability of housekeeping proteins in isolated adipocytes, examining their performance across technical replicates, dynamic range, and relative to TP. Unlike the earlier study in whole adipose tissue, we found that PARK7, ENOA, and FAH are not optimal normalization references for isolated mature human adipocytes. Specifically, PARK7 and ENOA exhibited higher CVs across technical replicates and a narrower dynamic range compared to commonly used housekeeping proteins, such as actin and tubulin. While FAH showed low variance in technical replicates, its limited dynamic range plateaued at only 10 µg of protein loading, suggesting that its low variance may reflect insensitivity to loading differences rather than reliability. Surprisingly, GAPDH, a widely used housekeeping protein, performed poorly in our experiments, showing high variance among technical replicates, a dynamic range that deviated from expected values, and significant variability between individuals.

In our subject cohort, TP levels varied by approximately 2-fold between individuals. This variation may be attributed to the high lipid content of adipocytes, which can cause contamination during protein isolation and interfere with BCA protein quantification. In contrast, densitometry data for housekeeping proteins showed a 4- to 10-fold difference between individuals, depending on the specific protein examined. While normalizing housekeeping proteins to TP helped reduce some of these discrepancies, significant differences persisted that could not be explained by loading errors. These findings underscore considerable variability in housekeeping protein expression, even among individuals with similar metabolic profiles.

Normalization choice also impacted the interpretation of GLUT4 expression. When GLUT4 was normalized to different references, the data patterns changed substantially, emphasizing the critical importance of using a consistent normalization reference when comparing results across blots or study cohorts. For normalization to be reliable, the reference must exhibit low variability both within and between individuals. However, factors such as the molecular weight of the target protein, laboratory preferences, and experimental design can make a single housekeeping protein unsuitable for all cases. Taken together, TP offers distinct advantages as a normalization reference. It does not rely on antibodies, is not constrained by molecular weight, and provides a broadly applicable approach regardless of the protein of interest. This makes TP an attractive and versatile choice for ensuring consistent and reliable comparisons across experiments.

One limitation of stain-free technology is that it works by modifying tryptophan residues, which could affect TP quantification depending on the sample's tryptophan content. While tryptophan metabolism varies between individuals and species, studies suggest that tryptophan levels in the proteome remain consistent within individuals of the same species [20,21]. Certain conditions, such as inflammation [22], cancer [23], kidney dysfunction [24], pregnancy [25], and treatments like interferon therapy [26] or indoleamine 2,3-dioxygenase 1 (IDO1) inhibitors [27], are associated with altered tryptophan metabolism. However, there is currently no evidence to suggest that these changes significantly impact the tryptophan content of cellular proteins or the abundance of tryptophan-containing proteins. In cases where changes in tryptophan-containing proteins are known or suspected, alternative TP normalization methods, such as Amido Black or

Ponceau Red staining (which target proteins via general electrostatic and hydrophobic interactions), or the use of stable housekeeping proteins (Table 1), may provide more reliable results.

Although tubulin outperformed TP in the presence of BSA, the fundamental issues with using housekeeping proteins as normalization references, highlighted in this study, remain. For optimal western blot normalization, it is preferable to eliminate exogenous protein contamination, such as BSA, rather than relying on housekeeping proteins as an alternative. In our experiments, excluding BSA from the adipocyte isolation protocol had no effect on adipocyte yield, however validation of removing BSA for downstream applications is needed. For experiments where the use of BSA is unavoidable, actin, as identified in this study, may serve as a more suitable normalization reference. However, whenever possible, adopting BSA-free protocols is strongly recommended to fully leverage the superior normalization capabilities of TP.

To our knowledge, this is the first study comparing TP and housekeeping protein normalization strategies in *isolated* primary human adipocytes. The substantial inter-individual variation in housekeeping protein expression observed in our study aligns with findings from other primary human tissue analyses: Similar variability has been reported in the human cortex [6], where actin displayed robust heterogeneity between individuals, while Ponceau TP staining provided more consistent normalization. In skeletal muscle, both GAPDH and actin demonstrated age-dependent expression decreases, while stain-free TP measurement remained stable for both inter- and intraindividual variation [8]. Similarly, placental reference proteins exhibited significant variation while total protein remained stable [9]. The biological basis for these differences likely stems from the dynamic nature of cellular metabolism. For example, metabolic enzymes such as GAPDH may vary with energy status [28,29], while structural proteins like actin could fluctuate with cell or tissue remodeling processes [30,31]. These variations are particularly relevant in adipocytes, where protein expression can be influenced by factors such as adipose depot location, BMI, age or systemic factors. The consistency of these findings across multiple primary human tissues underscores the limitations of housekeeping protein normalization and supports the broader applicability of TP normalization in human tissue analysis.

In summary, selecting an appropriate normalization strategy is essential for obtaining reliable and precise western blotting results. This study compared the normalization of GLUT4 to TP and housekeeping proteins, highlighting the significant impact of inter-individual variability in housekeeping protein expression on normalization outcomes.

While this study employed stain-free imaging to obtain TP densitometry data, alternative methods such as Coomassie Brilliant Blue, Amido Black, and Ponceau Red staining provide viable options for total protein quantification. These methods, including stain-free imaging, are generally more cost-effective than using antibodies for housekeeping proteins and preserve blot space, enabling greater multiplexing of proteins of interest.

In conclusion, we recommend total protein normalization as the gold standard for western blotting in studies involving primary mature human adipocytes, owing to its superior reliability and versatility.

## Supporting information

**S1 Table. Step-by-step sum normalization calculations for technical replicates and dynamic range.**
(XLSX)

**S2 Table. Protein concentrations and loading amount for each BSA control experimental condition for Fig 1.**
(DOCX)

**S3 Table. Clinical parameters for cohort used in Fig 4.**
(DOCX)

**S4 Table. Clinical parameters for all samples.**
(DOCX)

**S1 Fig.  Normalization schematic for technical replicates and dynamic range.** (A) Schematic for sum normalization of technical replicates, (B) theoretical and measured data, (C) fraction of sum, (D) divided by expected. (E) Schematic for sum normalization of dynamic range, (F) theoretical and measured data, (G) fraction of sum, (H) divided by expected.
(TIF)

**S2 Fig.  Simulation of western blot Data via seeded random number generation.** (A) Simulated raw signal intensities of a protein of interest (POI) and three different normalization references, each with a coefficient of variation (CV) of 5%, 15%, or 30%. (B) Normalized POI signal obtained by dividing the POI values by each respective normalization reference shown in (A). This demonstrates how increasing variability in the reference impacts the stability of the normalized signal, despite the POI having identical underlying variability in all cases.
(TIF)

**S3 Fig.  BSA-contaminated samples and non-specific signal.** (A) Protein concentrations, determined with a BSA standard curve and absorption measurement using a spectrophotometer with duplicates for all conditions. (B) GLUT4 stain, shown with increased exposure to better demonstrate specific GLUT4 staining in the BSA free samples.
(TIF)

**S4 Fig.  Technical replicates from OBHI individuals.** Coefficient of variation (CV) for total protein (TP) and respective housekeeping proteins, normalized to the sum (left panels), and for GLUT4 normalized to TP or housekeeping proteins (right panels). Each dot represents one well – a technical replicate of the same lysate sample run in separate lanes. Protein isolated from subcutaneous (SC) or omental (OM) adipocytes. Total protein demonstrated the lowest CV across all technical replicate blots, except when compared to FAH in the OM sample (A–F). When normalizing GLUT4, total protein showed the lowest CV in three blots (A, B and E), while actin (C), ENOA (D) and FAH and tubulin (F) exhibited lower CVs in three blots.
(TIF)

**S5 Fig.**   Technical replicate blots from Fig 2 and S3 Fig. (A) Subcutaneous obese normoinsulinemic (SC OBNI), total protein (TP), PARK7, GAPDH and GLUT4, (B) SC OBNI, TP, ENOA, actin and GLUT4, (C) SC OBNI, TP, FAH, tubulin and GLUT4, (D) omental (OM) OBNI, TP, PARK7, GAPDH and GLUT4, (E) OM OBNI, TP, ENOA, actin and GLUT4, (F) OM OBNI, TP, FAH, tubulin and GLUT4. Technical replicate blots from S3 Fig: (G) Subcutaneous obese hyperinsulinemic (SC OBHI), TP, PARK7, GAPDH and GLUT4, (H) SC OBHI, TP, ENOA, actin and GLUT4, (I) SC OBHI, TP, FAH, tubulin and GLUT4, (J) OM OBHI, TP, PARK7, GAPDH and GLUT4, (K) OM OBHI, TP, ENOA, actin and GLUT4, (L) OM OBHI, TP, FAH, tubulin and GLUT4.
(TIF)

**S6 Fig.  Dynamic range blots from** Fig 3. omental obese hyperinsulinemic (OM OBHI) lysate loaded on a gradient in 2.5 µg increments (2.5–20 µg). (A) Stain-free image of total protein (TP) and proteins stains for GLUT4, PARK7 and GAPDH, (B) GLUT4, ENOA and Actin, (C) GLUT4, FAH and tubulin.
(TIF)

**S7 Fig.  Cohort blots from Fig 4.** (A) Stain-free image of total protein (TP) and GLUT4, PARK7 and GAPDH staining. (B) Stain-free image of TP and GLUT4, ENOA and actin staining. (C) Stain-free image of TP and GLUT4, FAH and tubulin staining.
(TIF)

**S1 Raw images.  All original western blot images included in the manuscript, uncropped and unedited.**
(PDF)

## Acknowledgments

We acknowledge the assistance of staff at Ersta Hospital CRC, research nurse Charlotte Edberg (Danderyd Hospital) and laboratory technician Lena Appelsved for her assistance in processing human biopsy samples.

## Author contributions

**Conceptualization:** Leo J.S Westerberg, Benjamin Dedic, Kirsty Spalding.

**Formal analysis:** Leo J.S Westerberg, Benjamin Dedic.

**Funding acquisition:** Erik Näslund, Anders Thorell, Kirsty Spalding.

**Investigation:** Leo J.S Westerberg, Benjamin Dedic.

**Methodology:** Leo J.S Westerberg, Benjamin Dedic.

**Project administration:** Kirsty Spalding.

**Supervision:** Kirsty Spalding.

**Validation:** Leo J.S Westerberg, Benjamin Dedic.

**Visualization:** Leo J.S Westerberg, Benjamin Dedic.

**Writing – original draft:** Leo J.S Westerberg, Benjamin Dedic.

**Writing – review & editing:** Leo J.S Westerberg, Benjamin Dedic, Erik Näslund, Anders Thorell, Kirsty Spalding.

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
