## [Decision Letter · Decision Letter 0]

PONE-D-24-50811

Superior normalization using total protein for western blot analysis of human adipocytes

PLOS ONE

Dear Dr. Kirsty Spalding,

Thank you for submitting your manuscript to PLOS ONE. After careful consideration, we have decided that your manuscript does not meet our criteria for publication and must therefore be rejected.

Specifically:

I regret to inform you that your manuscript does not have the scientific significance required by this journal and we must therefore reject it.

I am sorry that we cannot be more positive on this occasion, but hope that you appreciate the reasons for this decision.

For your information and guidance, any specific comments explaining why I have reached this decision and those received from reviewers, if available, are listed at the end of this letter.

Kind regards,

Dr. V. V. Sathibabu Uddandrao

Academic Editor

PLOS ONE

Reviewer's Responses to Questions

**Comments to the Author**

1. Is the manuscript technically sound, and do the data support the conclusions?

Reviewer #1: Yes

Reviewer #2: Yes

Reviewer #3: No

2. Has the statistical analysis been performed appropriately and rigorously?

Reviewer #1: I Don't Know

Reviewer #2: Yes

Reviewer #3: No

3. Have the authors made all data underlying the findings in their manuscript fully available?

Reviewer #1: Yes

Reviewer #2: Yes

Reviewer #3: Yes

4. Is the manuscript presented in an intelligible fashion and written in standard English?

Reviewer #1: Yes

Reviewer #2: Yes

Reviewer #3: Yes

Reviewer #1: The introduction effectively sets up the study's objective, emphasizing the need for improved normalization strategies in Western blot analysis. Consider briefly addressing why TP normalization has not yet become a universal standard, despite its advantages, to contextualize the significance of this research.

The methods section is comprehensive, but some procedural descriptions, such as the justification for excluding BSA in adipocyte isolation, could benefit from further elaboration. For example, explain how the adjusted protocol impacts the broader applicability of the findings.

In the protein extraction and quantification section, it might help to clarify why the BCA assay was chosen over other protein quantification methods.

The results are thorough, with clear visualizations supporting the text. However, Figure legends could provide more contextual explanations of key observations for readers unfamiliar with the technical nuances of Western blot normalization.

Some variability in TP and housekeeping proteins across patient samples is discussed, but it would be helpful to hypothesize more concretely on the biological factors contributing to these differences.

Consider summarizing the key advantages of TP normalization over individual housekeeping proteins in a concise table or schematic for clarity.

The discussion appropriately highlights the benefits of TP normalization. It would strengthen the manuscript to compare these findings more explicitly with studies employing other normalization methods in similar contexts (e.g., specific adipocyte studies).

Addressing the potential limitations of TP normalization due to tissue-specific tryptophan content is valuable. Consider discussing whether this limitation could extend to clinical applications or other experimental designs.

The supplementary figures are well-constructed, but it may be helpful to integrate key supplementary findings directly into the main figures for a more cohesive narrative.

Ensure all figure captions are self-contained, explaining abbreviations and experimental conditions in detail.

The conclusion strongly advocates for TP normalization but could also acknowledge scenarios where housekeeping proteins might still be appropriate, particularly for specific experimental designs.

Ensure consistency in terminology and abbreviations throughout the manuscript.

Revisit the statistical analysis details to ensure they are sufficiently described, particularly regarding how variability was quantified across replicates and normalization references.

The ethics statement is clear, but more details on the demographic and clinical background of the patient cohort might enhance reproducibility and relevance.

Reviewer #2: The authors in their studies clearly revealed that TP (Total Protein) normalization methods in western blotting are far superior to housekeeping (GLUT4), actin, and tubulin protein-based normalization methods, including BSA contaminations.

The methods that they suggested are more reliable and cost-effective, as they are not using any antibodies in their work and are strain-free too.

But studies were limited to only primary mature human adipocytes. No more comparisons/discussions with with other cell types of human samples.

Noise eliminations and contaminations, like with BSA proteins, were discussed exhaustively, but no data was not shown for noise elimination or contamination removal.

Reviewer #3: Westerberg et al. have produced a manuscript investigating two approaches to Western blot normalization, housekeeping protein normalization and total protein normalization, in adipocytes.

The concept of the manuscript is not particularly novel, as stain-free total protein normalization has been a common way of normalizing Western blot data for more than ten years. However, it could still be of particular interest if done rigorously and in a way where a generalizable improvement can be shown in adipocytes. In this study, there are, unfortunately, too many issues with the experimental design.

i) The study relies on commercially available antibodies, and it is of utmost significance to ensure that the antibodies used in the experiments are thoroughly validated for the type of study performed. There is no discussion regarding the validation of the antibodies used, and there is only accessible validation data for the antibody targeting FAH (#PA5-42049). Additionally, this antibody is referred to as an anti-FAA antibody even though the target referenced by the vendors is FAH. There is no common gene abbreviation called FAA. This antibody is also disregarded due to signal saturation, which could easily be avoided by protocol optimization. A few recommended readings on antibody validation are:

- “Antibody validation” by Bordeaux et al. from 2010, BioTechniques.

- “Enhanced validation of antibodies for research applications” by Edfors et al. from 2018, Nature Communications.

- “Antibodypedia, a Portal for Sharing Antibody and Antigen Validation Data” by Björling and Uhlén from 2008, Molecular & Cellular Proteomics.

ii) The authors only investigate the normalization in the context of one protein target. The lack of different protein targets subjected to normalization nullifies any generalization of the findings, and making any generalized claims without investigating at least a smaller data set of ten different target proteins would be ill-advised.

iii) The manuscript does not have a proper statistical evaluation that supports the claim of improved accuracy with total protein normalization, and there are no established reference values for the samples that can indicate if the observed intensities actually reflect the truth.

**Do you want your identity to be public for this peer review?** For information about this choice, including consent withdrawal, please see our Privacy Policy

Reviewer #1: **No**

Reviewer #2: No

Reviewer #3: No

- - - - -

---

## [Author Response · Author response to Decision Letter 1]

30 Jan 2025

The response to reviewers includes figures and information that loses format if pasted into this space. Instead the response to reviewers has been uploaded as PDF file 'Response to reviewers.pdf'.

---

## [Decision Letter · Decision Letter 1]

Dear Dr. Spalding,

Thank you for submitting your manuscript to PLOS ONE. After careful consideration, we feel that it has merit but does not fully meet PLOS ONE’s publication criteria as it currently stands. Therefore, we invite you to submit a revised version of the manuscript that addresses the points raised during the review process.

We look forward to receiving your revised manuscript.

Kind regards,

Jérôme Robert, PhD

Academic Editor

PLOS ONE

Journal Requirements:

5. We note that your Data Availability Statement is currently as follows: [All relevant data are within the manuscript and its Supporting Information files.]

If there are ethical or legal restrictions on sharing a de-identified data set, please explain them in detail (e.g., data contain potentially sensitive information, data are owned by a third-party organization, etc.) and who has imposed them (e.g., an ethics committee). Please also provide contact information for a data access committee, ethics committee, or other institutional body to which data requests may be sent. If data are owned by a third party, please indicate how others may request data access."

Additional Editor Comments (if provided):

Reviewers' comments:

Reviewer's Responses to Questions

**Comments to the Author**

Reviewer #1: All comments have been addressed

Reviewer #3: (No Response)

Reviewer #4: (No Response)

2. Is the manuscript technically sound, and do the data support the conclusions?

Reviewer #1: Yes

Reviewer #3: No

Reviewer #4: Yes

3. Has the statistical analysis been performed appropriately and rigorously?

Reviewer #1: Yes

Reviewer #3: No

Reviewer #4: Yes

4. Have the authors made all data underlying the findings in their manuscript fully available?

Reviewer #1: Yes

Reviewer #3: Yes

Reviewer #4: Yes

5. Is the manuscript presented in an intelligible fashion and written in standard English?

Reviewer #1: Yes

Reviewer #3: Yes

Reviewer #4: Yes

Reviewer #1: I have gone through with the revised manuscript and found it suitable for the publication. Authors have addressed all comments

Reviewer #3: The authors seem to have misunderstood the comment regarding assessing normalization in multiple target proteins. In the study, normalization is only evaluated in the context of GLUT4. To claim that the normalization is superior for experiments conducted in primary human adipocytes, the normalization strategies must be assessed for multiple proteins of variable expression between individuals. Without this comparison, the claims of superior normalization can only be applied to GLUT4 specifically.

Additionally, there is a significant need to discuss the validation of the antibodies used beyond “all commercially validated” when using these as the basis for comparing different normalization strategies. As stated in the previous reviewer comments, there is only publicly available validation data for one of the antibodies, and a discussion and justification for the choice of these antibodies should be included in the manuscript.

As the authors state in their response, they have done a robust statistical assessment of the measurement stability. Still, stability is not the same thing as accuracy. Without any established reference values for GLUT4 in the different samples from individuals (preferably with a different methodology), there is no evidence of improved accuracy in the measurements. Accuracy is a measurement of how well the quantification reflects the actual amount of the analyte. Accuracy can, for example, be assessed by measuring the samples with an orthogonal method or by spiking a sample with different amounts of a recombinant protein. This is of significant importance when working with normalization strategies, as the normalization directly impacts the measurement values obtained from an analysis.

Reviewer #4: The manuscript entitled “Superior normalization using total protein for western blot analysis of human

adipocytes” demonstrates that total protein might be a better normalization element in WB quantification of protein lysates when studying adipose tissue and adipocytes. In general, the experiments are well designed and presented. I have few questions and comments that may improve the readability of the manuscript:

• Lines 246-253 “Overexposure of the blot showed that the expected GLUT4 signal (~54 kDa) was clearly present in the BSA-free samples while the BSA-isolated samples showed non-specific bands at ~60 kDa (S2B Fig). These results can be explained by anti-GLUT4 antibody binding non-specifically to BSA (65 kDa) leading to false positives in BSA-isolated samples but not in BSA-free samples. Thus, subsequent experiments within this study used BSA-free samples, accentuating the importance of limiting BSA, or other protein contaminants, in western blotting”. Is there a reason why the authors speculate that is the anti-GLUT4 that is cross-reacting with BSA and not the secondary antibody?

• I am curious if is there a technical/biological reason for GLUT4 selection as target gene? Notoriously, WB of membrane proteins are less clean and often characterized by double bands not very sharped associated to the glycosylated and non-glycosylated forms.

• Figure 2 shows technical replicates from OBNI individuals. Do I understand correctly that each dot represent different runs from the same lysate sample? If so, please add/reiterate this when you describe the results and in the figure legend.

• I do not see a callout for table 1.

• Sometimes μg was reported as ug (e.g., line 263, line 309). Please correct.

• Sometimes numbers were reported with comma (e.g., 2,5 – line 300), sometimes with dot (e.g., 27.3, line 320). I suggest using everywhere the dot format.

**Do you want your identity to be public for this peer review?** For information about this choice, including consent withdrawal, please see our Privacy Policy

Reviewer #1: **Yes: ** Umesh Kumar

Reviewer #3: No

Reviewer #4: No

---

## [Author Response · Author response to Decision Letter 2]

25 Jun 2025

Rebuttal PONE-D-24-50811R1

“Superior normalization using total protein for western blot analysis of human adipocytes”

June 18, 2025

Stockholm, Sweden

Dear Editor and Reviewers,

Thank you for taking the time to review our work and for your thoughtful feedback.

Please find below our point-by-point response to comments (author response in blue) as well as our response to the general comments requested of the reviewers.

All line references refer to the clean version of the manuscript.

Thank you again for this constructive review process, we believe the manuscript has been substantially strengthened as a result.

Kind regards,

Kirsty

Kirsty Spalding

Professor | Vice-Chair Department of Cell and Molecular Biology

Wallenberg Scholar

Karolinska Institutet

Biomedicum B7 | Solnavägen 9 | SE-171 65 Solna

Aula Medica | Nobels väg 6 | Floor 7 | SE-171 77 Stockholm

Phone: +46 (0)70 437 1542

E-mail: kirsty.spalding@ki.se

Karolinska Institutet – a medical university

Reviewer 1

I have gone through with the revised manuscript and found it suitable for the publication. Authors have addressed all comments.

We thank you for your positive evaluation and constructive comments, which we believe have helped improve the quality and clarity of our work.

Reviewer 3

1. The authors seem to have misunderstood the comment regarding assessing normalization in multiple target proteins. In the study, normalization is only evaluated in the context of GLUT4. To claim that the normalization is superior for experiments conducted in primary human adipocytes, the normalization strategies must be assessed for multiple proteins of variable expression between individuals. Without this comparison, the claims of superior normalization can only be applied to GLUT4 specifically.

Response to Reviewer 3, Comment 1

Thank you for this important point. We agree that demonstrating the generalisability of our normalization strategy beyond a single protein target strengthens our conclusions. To address this, we have:

(i) Clarified the role of GLUT4 as an illustrative example, not the sole basis for our claims.

(ii) Added a simulation-based analysis to show how reference-protein variability affects normalized measurements irrespective of the target protein.

We would like to clarify that GLUT4 was included merely as a “real-world” example of our workflow, not to imply that our conclusions apply only to GLUT4. In practice, the benefits of total-protein normalization, namely, its superior technical reproducibility, increased inter-individual stability, and broader dynamic range, are independent of the particular protein being measured.

To illustrate this universality, we generated simulated Western-blot datasets with controlled variability. Using the Box-Muller transformation, we created a “protein of interest” (POI) dataset with a realistic technical CV of 19.9% (mean = 1) and three randomized independent “reference” datasets with CVs of 5%, 15%, and 30%. We then normalized the POI values to each reference and recalculated CVs.

(i) Stable reference (5% CV): Normalization maintained high precision (final CV = 16.6%), closely matching the POI’s original variability.

(ii) Moderately variable reference (15% CV): Normalization inflated variability (final CV = 30.6%).

(iii) Highly variable reference (30% CV): Normalization further increased variability (final CV = 47%), a 2.8-fold loss of precision compared to the stable reference (16.6% stable vs 47% highly variable).

These results (now shown in the revised manuscript’s Methods, starting line 199; Results, starting line 210; and New S2 Fig.) demonstrate that reference-protein stability directly governs the precision of normalized measurements, regardless of which target protein is analyzed. Importantly, all simulations assume identical loading and isolate only the effects of reference variability on quantification precision. While biological factors (e.g., correlations between target and reference expression) are not accounted for in the model here, our findings establish a general analytical principle: total-protein normalization minimizes technical noise and maximizes reproducibility across any protein target.

New S2 Fig. Simulation of Western-blot normalization via seeded random data. (A) Raw signal intensities for a protein of interest (POI) and three normalization references with increasing coefficients of variation (CV = 5 %, 15 %, 30 %). (B) Normalized POI signals (POI/reference) for each CV level. As reference variability rises, normalized-signal stability deteriorates, even though the POI’s underlying variability remains constant.

2. Additionally, there is a significant need to discuss the validation of the antibodies used beyond “all commercially validated” when using these as the basis for comparing different normalization strategies. As stated in the previous reviewer comments, there is only publicly available validation data for one of the antibodies, and a discussion and justification for the choice of these antibodies should be included in the manuscript.

Response to Reviewer 3, Comment 2

We appreciate the reviewer’s emphasis on rigorous antibody validation. Below, we summarise both published evidence and our in-house checks for each antibody used (we are happy to include images from all referenced figures in our rebuttal if the editor/reviewer wishes):

GLUT4 antibody (66846-1-Ig, Proteintech)

• Literature support: Cited in >90 peer-reviewed studies for Western blot (WB) and immunocytochemistry (ICC) with consistent detection of the 45–55 kDa GLUT4 band (see list: https://www.citeab.com/antibodies/6638026-66846-1-ig-glut4-monoclonal-antibody?des=3de7421913174b6d).

• Manufacturer data: Raised against AA 1–100 of human GLUT4; minimal cross-reactivity to other GLUT isoforms (datasheet).

PARK7 antibody (ab18257, Abcam)

• Vendor validation: Abcam’s KO-cell analysis shows complete signal loss in PARK7 KO HAP1 cells (https://www.abcam.com/en-us/products/primary-antibodies/park7-dj1-antibody-ab18257).

• Cross-species: Validated on human, mouse, and rat lysates.

FAH antibody (PA5-42049, Thermo Fisher)

• Knockdown validation: Demonstrated in DOI:10.1002/mgg3.2090, Fig. 3A.

Housekeeping-protein antibodies

• GAPDH (12004167), Actin (12004163), Tubulin (12004165):

Literature usage: Each antibody has been used extensively (GAPDH: 25 studies; Actin: 50; Tubulin: 44, see https://www.citeab.com/…), consistently yielding single bands at expected weights.

Practical constraints: True KO/knockdown is often not feasible for essential proteins; instead, we rely on extensive community usage and concordant fractionation data (e.g., purity markers in cytosolic vs. nuclear fractions).

ENO1 antibody (MBS9231722, MyBioSource)

• Overexpression validation: Shown in Blood 2008 (10.1182/blood-2008-08-170837, Fig. 5A).

While we have selected antibodies with strong literature precedent, we recognise that no single antibody validation is perfect. This limitation further motivates our adoption of total-protein normalization: it minimises dependence on any one antibody’s specificity and provides a more robust, reproducible baseline across experiments.

3. As the authors state in their response, they have done a robust statistical assessment of the measurement stability. Still, stability is not the same thing as accuracy. Without any established reference values for GLUT4 in the different samples from individuals (preferably with a different methodology), there is no evidence of improved accuracy in the measurements. Accuracy is a measurement of how well the quantification reflects the actual amount of the analyte. Accuracy can, for example, be assessed by measuring the samples with an orthogonal method or by spiking a sample with different amounts of a recombinant protein. This is of significant importance when working with normalization strategies, as the normalization directly impacts the measurement values obtained from an analysis.

Response to Reviewer 3, Comment 3

Thank you for this insightful point. We apologise for any lack of clarity regarding the scope of our analysis. Our goal was not to determine the absolute abundance of GLUT4 in adipocyte lysates, but rather to use GLUT4 as a representative “real-world” example to illustrate how the choice of normalization reference, particularly different housekeeping proteins, can dramatically influence the precision of normalized protein measurements across individuals.

To make this explicit, we have:

1. Emphasised the illustrative nature of GLUT4

We have revised the text to state clearly that GLUT4 serves only as an example target. Our aim is to highlight variability in normalization references, not to benchmark GLUT4 quantification accuracy per se.

2. Linked back to our simulation analysis

As detailed in our response to Comment 1, we now include simulated Western-blot datasets showing that, even with identical underlying protein signals, increasing variability in the reference inflates normalized-signal imprecision. This controlled demonstration (New S2 Fig) underscores that reference stability, not the target protein, drives normalized-measurement precision.

3. Clarified Figure 4’s purpose

We have updated the legend (line 386) and main text to specify that Figure 4D-F illustrates inter-individual variability of six housekeeping proteins, before and after total-protein normalization. These panels are intended to show that:

- Normalizing housekeeping proteins to total protein only partly reduces their inter-patient variability.

- Different housekeeping proteins display distinct variability profiles across donors.

- Such variability violates the assumption that a normalization reference is stable and independent of biological conditions.

We agree that evaluating accuracy (i.e., closeness to the true protein amount) would require orthogonal methods or spike-in standards. While this is beyond the present study’s scope, our simulation and real-data analyses together establish a compelling case that total-protein normalization minimizes technical noise and yields more reproducible quantification, an essential prerequisite for accuracy in any downstream measurement. We have made these clarifications in the Methods section (line 193), the Results section (line 423), and the Discussion section (line 530) of the revised manuscript to ensure that the distinction between precision (measurement variability) and accuracy (true-value agreement) is fully transparent.

Reviewer 4:

1. Lines 246-253 “Overexposure of the blot showed that the expected GLUT4 signal (~54 kDa) was clearly present in the BSA-free samples while the BSA-isolated samples showed non-specific bands at ~60 kDa (S2B Fig). These results can be explained by anti-GLUT4 antibody binding non-specifically to BSA (65 kDa) leading to false positives in BSA-isolated samples but not in BSA-free samples. Thus, subsequent experiments within this study used BSA-free samples, accentuating the importance of limiting BSA, or other protein contaminants, in western blotting”. Is there a reason why the authors speculate that is the anti-GLUT4 that is cross-reacting with BSA and not the secondary antibody?

Response to Reviewer 4, Comment 1

Thank you for pointing this out. You are correct that we cannot definitively attribute the cross-reactivity to the primary versus the secondary antibody without additional experiments. To avoid overinterpretation, we have revised the text (Lines 279–285) as follows:

“Overexposure of the blot showed that the expected GLUT4 signal (~54 kDa) was clearly present in BSA-free samples, while BSA-isolated samples exhibited a non-specific band at ~60 kDa (S2B Fig). These results can be explained by non-specific antibody binding to BSA (65 kDa), leading to false positives in BSA-isolated samples but not in BSA-free samples. Thus, subsequent experiments within this study used BSA-free samples, accentuating the importance of limiting BSA, or other protein contaminants, in western blotting.”

This wording removes any implication about which antibody is responsible, accurately reflecting our observations. Thank you for helping us clarify this point.

2. I am curious if is there a technical/biological reason for GLUT4 selection as target gene? Notoriously, WB of membrane proteins are less clean and often characterised by double bands not very sharped associated to the glycosylated and non-glycosylated forms.

Response to Reviewer 4, Comment 2

Thank you for this thoughtful question. GLUT4 was chosen purely as a methodological test case because, in our hands, it yields robust, clear, and highly reproducible bands under our optimised western-blot protocol for mature human adipocytes. We acknowledge that membrane proteins often display glycosylation-related doublets or diffuse bands; however, GLUT4 consistently produces interpretable signals across multiple donor samples, making it well suited for evaluating normalization strategies.

Importantly, the intent was not to draw any biological conclusions about GLUT4 regulation, but rather to use a realistic adipocyte protein target to benchmark how different normalization approaches perform.

3. Figure 2 shows technical replicates from OBNI individuals. Do I understand correctly that each dot represent different runs from the same lysate sample? If so, please add/reiterate this when you describe the results and in the figure legend.

Response to Reviewer 4, Comment 3

Thank you for the suggestion. We have clarified both the figure legend and the Results text to make this explicit:

Figure 2 legend (line 316) now reads:

“Each dot represents one well - a technical replicate of the same lysate sample run in separate lanes.”

Results section (line 292) now changed to read: “First, the assessment involved technical replicates, where each well in a gel was loaded with 10 µg of protein from the same sample. Samples were obtained from four different individuals: two obese hyperinsulinemic (OBHI) patients (one subcutaneous (SC) and one omental (OM)) and two obese normoinsulinemic (OBNI) patients (one SC and one OM). Each sample was analyzed across three separate blots to assay all housekeeping proteins, allowing for the normalization of GLUT4 levels to two housekeeping proteins and TP within each blot.”

We have made the analogous update to S4 Fig legend (line 655) for consistency.

4. I do not see a callout for table 1.

Response to Reviewer 4, Comment 4

Table 1 is cited twice in the current version of the manuscript: first on line 427 and again on line 498. It is initially referenced in the final paragraph of the Results section—not to present new data, but to provide a summary of the advantages and disadvantages of different normalization references across various experimental designs. We acknowledge that, because the table is introduced relatively late and does not contain original data, the callout may be easily overlooked. However, we believe that its current placement is appropriate and serves its intended purpose within the manuscript structure.

5. Sometimes μg was reported as ug (e.g., line 263, line 309). Please correct.

Response to Reviewer 4, Comment 5

We appreciate your attention to detail and have corrected all instances of “ug” to “µg” throughout the manuscript.

6. Sometimes numbers were reported with comma (e.g., 2,5 – line 300), sometimes with dot (e.g., 27.3, line 320). I suggest using everywhere the dot format.

Response to Reviewer 4, Comment 6

We appreciate the reviewer’s attention to consistency and have replaced all instances of numbers reported with a comma to the dot format throughout the manuscript.

General questions from the journal:

1. If the authors have adequately addressed your comments raised in a previous round of review and you feel that this manuscript is now acceptable for publication, you may indicate that here to bypass the “Comments to the Author” section, enter your conflict of interest statement in the “Confidential to Editor” section, and submit your "Acce

---

## [Editor Report · Decision Letter 2]

Superior normalization using total protein for western blot analysis of human adipocytes

PONE-D-24-50811R2

Dear Dr. Spalding,

We’re pleased to inform you that your manuscript has been judged scientifically suitable for publication and will be formally accepted for publication once it meets all outstanding technical requirements.

Kind regards,

Jérôme Robert, PhD

Academic Editor

PLOS ONE
---

## [Editor Report · Acceptance letter]

PONE-D-24-50811R2

PLOS ONE

Dear Dr. Spalding,

I'm pleased to inform you that your manuscript has been deemed suitable for publication in PLOS ONE. Congratulations! Your manuscript is now being handed over to our production team.

Kind regards,

on behalf of

Dr. Jérôme Robert

Academic Editor

PLOS ONE